# Estimated incidence and case fatality rate of traumatic brain injury among children (0–18 years) in Sub-Saharan Africa. A systematic review and meta-analysis

**Martin Ackah[1,2]\*, Mohammed Gazali Salifu[2,3], Cynthia Osei Yeboah[1]**

**1** Department of Physiotherapy, Korle Bu Teaching Hospital, Accra, Ghana, **2** Department of Epidemiology and Disease Control, School of Public Health, University of Ghana, Accra, Ghana, **3** Policy Planning Budgeting Monitoring and Evaluation Directorate, Ministry of Health, Accra, Ghana

\* martinackah10@gmail.com

**Data Availability Statement:** All relevant data are within the paper and its Supporting information files.

## Abstract

### Introduction

Studies from Sub-Saharan Africa (SSA) countries have reported on the incidence and case fatality rate of children with Traumatic Brain Injury (TBI). However, there is lack of a general epidemiologic description of the phenomenon in this sub-region underpinning the need for an accurate and reliable estimate of incidence and outcome of children (0–18 years) with TBI. This study therefore, extensively reviewed data to reliably estimate incidence, case fatality rate of children with TBI and its mechanism of injury in SSA.

### Methods

Electronic databases were systematically searched in English via Medline (PubMed), Google Scholar, and Africa Journal Online (AJOL). Two independent authors performed an initial screening of studies based on the details found in their titles and abstracts. Studies were assessed for quality/risk of bias using the modified Newcastle-Ottawa Scale (NOS). The pooled case fatality rate and incidence were estimated using DerSimonian and Laird random-effects model (REM). A sub-group and sensitivity analyses were performed. Publication bias was checked by the funnel plot and Egger's test. Furthermore, trim and fill analysis was used to adjust for publication bias using Duval and Tweedie's method.

### Results

Thirteen (13) hospital-based articles involving a total of 40685 participants met the inclusion criteria. The pooled case fatality rate for all the included studies in SSA was 8.0%; [95% CI: 3.0%-13.0%], and the approximate case fatality rate was adjusted to 8.2%, [95% CI:3.4%-13.0%], after the trim-and-fill analysis was used to correct for publication bias. A sub-group analysis of sub-region revealed that case fatality rate was 8% [95% CI: 2.0%-13.0%] in East Africa, 1.0% [95% CI: 0.1% -3.0%] in Southern Africa and 18.0% [95% CI: 6.0%-29.0%] in

**Funding:** The authors received no specific funding for this work.

**Competing interests:** The authors have declared that no competing interests exist.

west Africa. The pooled incidence proportion of TBI was 18% [95% CI: 2.0%-33.0%]. The current review showed that Road Traffic Accident (RTA) was the predominant cause of children's TBI in SSA. It ranged from 19.1% in South Africa to 79.1% in Togo.

## Conclusion

TBI affects 18% of children aged 0 to 18 years, with almost one-tenth dying in SSA. The most common causes of TBI among this population in SSA were RTA and falls. TBI incidence and case fatality rate of people aged 0–18 years could be significantly reduced if novel policies focusing on reducing RTA and falls are introduced and implemented in SSA.

## Introduction

Traumatic Brain Injury (TBI) in children is acquired brain injury following trauma, and is similar to those of adults but differs in both management and pathophysiology [1]. The variations are due to age-related anatomical changes, injury mechanisms depending on the child's physical capacity, and the complexity of evaluating pediatric populations neurologically [1].

TBI annually affects 64 to 74 million people worldwide from all causes, and accounts for 11% of overall global disability years [2–4]. It accounts for a large proportion of childhood deaths in Europe [5, 6] and leading cause of mortality and morbidity in Low and Middle Income Countries (LMICs) [7, 8]. The burden of trauma and associated TBI is significantly higher in Low And Middle-Income Countries (LMICs), despite the fact that the incidence of pediatric with TBI differs widely [9, 10].

Berger et al. estimated that only 65% of children with severe TBI survive [11, 12]. The outcome of brain injury is very detrimental to the child, family and by extension the country. For instance, studies have found that TBI can lead to long-term cognitive and neurobehavioral deficiencies, as well as intellectual, academic, and personality adjustment issues, and familial stress [13, 14]. This could lead to a reduction in future capabilities or outright dependency in adulthood, both of which are contributing factors to poverty.

Dewan and colleagues discovered that road traffic crashes and falls accounted for the majority of injuries in the pediatric population in their global TBI study [10]. A UK study observed that falls account for 60% of TBIs in children < 5 years whilst RTAs led with 37% within the age group of 10–15 years [15]. In the same vein a population-based study in France reported RTA as the commonest followed by falls in all age group [16]. Additionally, another study conducted in Sub-Sahara Africa (SSA) identified RTA as the common cause of pediatric neurotrauma in all age groups [17].

Furthermore, studies from SSA countries have reported on the incidence and case fatality rate of children with TBI. However, there is lack of a general epidemiologic description of the phenomenon in this sub-region underpinning the need for an accurate and reliable estimate of the incidence and outcome of TBI in children, as a result, a well-organized systematic review and meta-analytic models are required.

This study therefore extensively reviewed data to reliably estimate the incidence, case fatality rate of children with TBI and its mechanism of injury in SSA. This could lead to better preventive measures, treatment, and outcomes.

## Methods

### Protocol registration

The present protocol has been registered with International Prospective Register of Systematic Reviews (PROSPERO) database with registration number CRD42021248726, and reported in compliance with Preferred Reporting Items for Systematic review and Meta-analyses (PRISMA) checklist [18] [S1 Table].

### Criteria for considering studies in the review

**Types of studies.**   Prospective or retrospective hospital-based studies published between 2000 and 2020 which reported children with TBI in SSA were considered for inclusion. Animal studies, reviews, commentaries, and letter to the editor were excluded.

**Setting/Participants.**   Studies from SSA countries reporting TBI in children. The review included children aged between 0–18 years.

**Type of intervention.**   Studies reporting on the incidence or case fatality rate of TBI involving children in SSA.

**Outcome of interest.**   The primary outcome of interest is the estimated incidence and case fatality rate of pediatrics' TBI in SSA. The secondary outcome was the mechanism of injury associated with pediatrics' TBI.

### Data sources and search strategies

Electronic databases were systematically searched in English via Medline (PubMed), Google Scholar, and Africa Journal Online (AJOL). The search was limited to January, 2000- December, 2020. Additional relevant articles were hand-searched in the reference lists of all included studies. Grey literature was conducted via google. Keywords such as "pediatric", "childhood," "traumatic brain injury,", "traumatic head injury," "mortality rate," "case fatality rate," "death rate," "incidence," "burden," "Sub-Saharan Africa". 'The Boolean operators "OR" and "AND" were used to combine these keywords. The search strategy is shown in S2 Table.

### Screening and selecting studies

Two independent authors (MA and MGS) performed an initial screening of studies based on the details found in their titles and abstracts. The same independent investigators performed the full-paper screening. Disagreements were resolved by consensus. To ensure that independent reviewers apply the selection criteria consistently, a screening guide was used [19].

### Data extraction and management

Data were extracted using a pre-tested and standardized excel spreadsheet. Data such as the last name of the first author, year of publication, country, type of study, sample size, sex, incidence, case fatality, age range, duration of study, severity measure, and mechanism of injury were extracted. The articles were managed with Mendeley referencing manager.

### Risk of bias assessment

Studies in the systematic review and meta-analysis were assessed for quality/risk of bias using the modified Newcastle-Ottawa scale (NOS) [20]. Two independent reviewers (MA and MGS) completed the process, with the average serving as the study's final score. The inter-rater reliability was 0.9 [kappa = 0.9]. The assessment tool contains three domains; methodological quality, comparability of the study and outcome measure and related statistical analysis and

are scored on a 'star' system [20]. Furthermore, the review rated the overall quality of the studies into three categories; [low risk of bias (score7-10), moderate risk of bias (score;5–6), and high risk of bias (socre;0–4)].

## Statistical analyses

Extracted data were exported into Stata (version 16; Stata Cooperation, TX, USA) from Microsoft excel 2013 for all analyses. The descriptive findings were presented and summarized in Tables. The pooled case fatality rate was estimated using DerSimonian and Laird random-effects model (REM) at 95% confidence interval as well as the incidence proportion of TBI. Heterogeneity was assessed by the $I^2$ and Q statistics and defined as ($I^2 > 50\%$, $p < 0.05$) indicating a substantial heterogeneity [21]. A sub-group analysis was performed based on sub-region (West Africa vs. East Africa vs. Southern Africa), publication year (<2017 and >2017), study design (Prospective vs. Retrospective), and quality score (low risk vs. moderate risk vs. high risk of bias) to determine possible source of heterogeneity. Leave one out sensitivity analysis was performed to examine the effects of a single study on the overall pooled estimate. Publication bias was checked by the funnel plot and Egger's test. Furthermore, trim and fill analysis was used to adjust for publication bias using Duval and Tweedie's method [22].

## Results

### Study selection

Electronic database searches in Medline (PubMed), Google Scholar, and AJOL yielded a total of 820 records. After excluding duplicates, 200 articles were eligible. Thirty (30) complete articles were evaluated for eligibility and 13 papers (n = 40685) [4, 17, 12, 23–32] met the inclusion criteria and were included in the final qualitative and meta-analysis. However, 4 studies [26, 29–31] were included in the meta-analysis for the pooled incidence (Fig 1).

### Study characteristics

Table 1 show the characteristics of the included studies. Out of the 13 studies included, 69% were retrospectively designed. The sample size ranged from 91 to 37,610 with estimated participants of 40,685. The studies were published between 2004 and 2020. Four of the included studies were conducted in Eastern Africa, 4 in Southern Africa and 5 in Western Africa. The current review showed that RTA was the predominant cause of children's TBI in SSA. Fall was the second commonest mechanism of injury in SSA. This also ranged from 5.1% in South Africa [12], and 41.2% in south Africa [26]. The common outcome measure for severity was the Glasgow Coma Scale (GCS). TBI affects male children more often than females in SSA.

### Pooled case fatality rate of traumatic brain injury among children in Sub-Saharan Africa

In the meta-analysis, the pooled case fatality rate for all the included studies in SSA was 8.0%; [95% CI: 3.0%-13.0%]. A significant heterogeneity was detected across the included studies ($I^2$ = 64.8%, p<0.000) (Fig 2).

A sub-group analysis of sub-region revealed that case fatality rate was 8% [95% CI: 2.0%-13.0%] in East Africa, 1.0% [95% CI: 0.1% -3.0%] in Southern Africa and 18.0% [95% CI: 6.0%-29.0%] in West Africa. Similarly, quality score sub-group analysis showed that case fatality rate for low, moderate, and high risk of bias studies were 8.0%, [95% CI: 2.0%-14.0%], 9.0%, [95% CI: 2.0%-17.0%], and 15.0%, [95% CI: 2.0%-31.0%] respectively. Studies that were published before 2017 had a pooled case fatality rate of 9.0%, [95% CI: 2.0%-19.0%] as

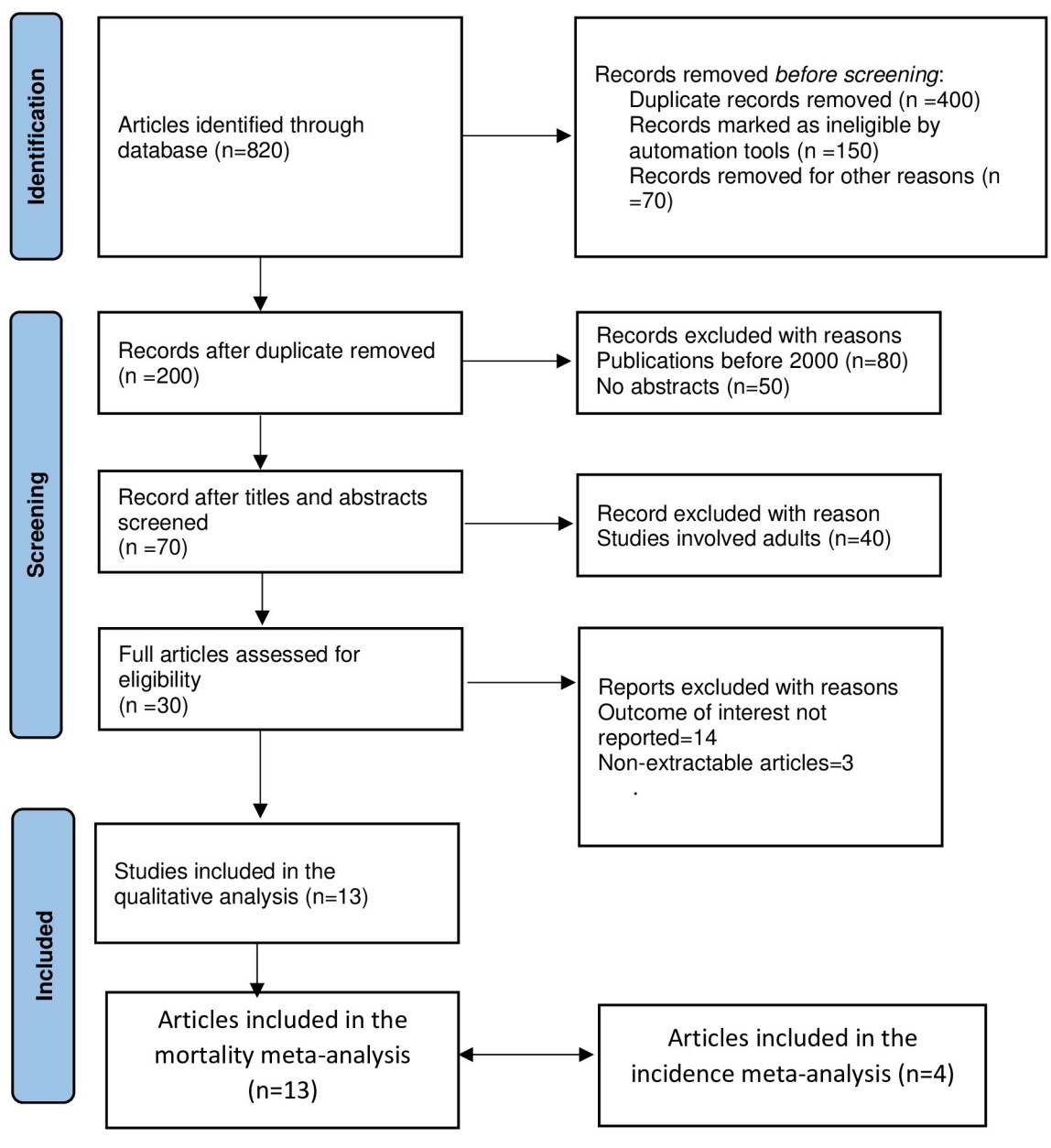

**Fig 1. PRISMA flowchart diagram of study selection.**

compared to studies from 2017 and above 10.0%, [95% CI: 5.0%-14.0%]. Based on the study design, prospective study and retrospective had a pooled case fatality rate of 7.0%, [95% CI: 0.0%-13.0%] and 10.0%, [95% CI: 3.0%-17.0%] respectively (Fig 3).

## Pooled incidence proportion of children with traumatic brain injury in Sub-Saharan Africa

Four studies reported on the incidence of children's TBI in SSA. The pooled analysis indicated incidence proportion of 18% [95% CI: 2.0%-33.0%]. A substantial heterogeneity ($I^2$ = 98.9%, P<0.000) was seen among the studies (Fig 4).

**Table 1. Characteristics of studies included in the review.**

| Author | Country | Study Design | Setting | Duration | Age Range (years) | Male: Female Ratio | Sample Size | Case Fatality | Admission GCS [%] | | | Missing | Mechanism of Injuries [%] | | | |
|---|---|---|---|---|---|---|---|---|---|---|---|---|---|---|---|---|
| | | | | | | | | | Mild (13–15) | Moderate (9–12) | Severe (<9) | | RTA | Falls | Intentional | Others |
| Abdelgadir et al. [4] | Uganda | Retrospective | Referral Hospital | 2012–2015 | 0–18 | 1.6:1 | 381 | 38 | 53.8 | 29.8 | 16.4 | 0 | 71.1 | 11.5 | 9.9 | 7.6 |
| Vaca et al. [23] | Uganda | Prospective | Referral Hospital | 2014–2015 | 0–17 | 2.0:1 | 347 | 34 | 46 | 32 | 17 | 4 | 72 | 9 | 12 | 7 |
| Punchak et al. [17] | Uganda | Prospective | Referral Hospital | 2016–2017 | 0–18 | 1.6:1 | 100 | 4 | 55 | 30 | 11 | 3 | 75 | 6 | 10 | 7 |
| Schrieff et al. [12] | South Africa | Retrospective | University/ Specialist Hospital | 2000–2011 | 0–15 | 1.9:1 | 137 | 20 | Not reported | Not reported | Not reported | | 75.9 | 5.1 | 6.6 | 3.7 |
| Bedry et al. [30] | Ethiopia | Prospective | University/ Specialist Hospital | 2017–2018 | 7m–14 | 2.2:1 | 317 | 10 | 72.9 | 19.2 | 7.9 | 0 | 45.4 | 32.8 | 12.6 | 8.8 |
| Udoh et al. [29] | Nigeria | Prospective | Teaching/ Referral Hospital | 2006–2011 | 3m–17 | 1.1:1 | 127 | 11 | 29.1 | 30.7 | 40.2 | 0 | 67.7 | 15 | 5.2 | 1.6 |
| Buitendag et al. [24] | South Africa | Retrospective | Prospective Digital Registry | 2012–2016 | ≤18 | 2.4:1 | 563 | 11 | 80.1 | 11.9 | 8 | 0 | 43 | 18 | 19 | 20 |
| Okyere-Dede et al. [28] | South Africa | Retrospective | Tertiary Hospital | 1999–2001 | 0–15 | 2.0:1 | 506 | 18 | 80.1 | 10.3 | 9.6 | 0 | 63 | 23 | 5 | 8 |
| Lalloo et al. [26] | south Africa | Retrospective | University/ Specialist Hospital | 1991–2001 | 0–13 | 1.4:1 | 37610 | 75 | Not reported | Not reported | Not reported | | 19.1 | 41.2 | 13.1 | 31.5 |
| Egbonhou et al. [31] | Togo | Retrospective | University Hospital | 2012–2018 | 0–15 | 2.0:1 | 91 | 29 | 52.7 | 39.6 | 7.7 | 0 | 79.1 | 19.8 | | 1.1 |
| Hode et al. [32] | Benin | Retrospective | University Hospital | 2012–2013 | 0–16 | 1.4:1 | 102 | 4 | 51.9 | 33.3 | 14.8 | 0 | 62.8 | | | |
| Kouitcheu et al. [25] | Cote D'voire | Retrospective | University Hospital | 2000–2017 | <16 | 1.8:1 | 292 | 39 | 53.8 | 36.8 | 9.4 | 0 | 78.7 | 9.4 | 2.6 | 7.3 |
| Mendy et al. [27] | Senegal | Retrospective | General Hospital | 2000–2010 | 0–15 | | 112 | 39 | Not reported | Not reported | Not reported | | 74.9 | | | |

RTA = Road Traffic Accident, GCS = Glasgow Coma Scale.

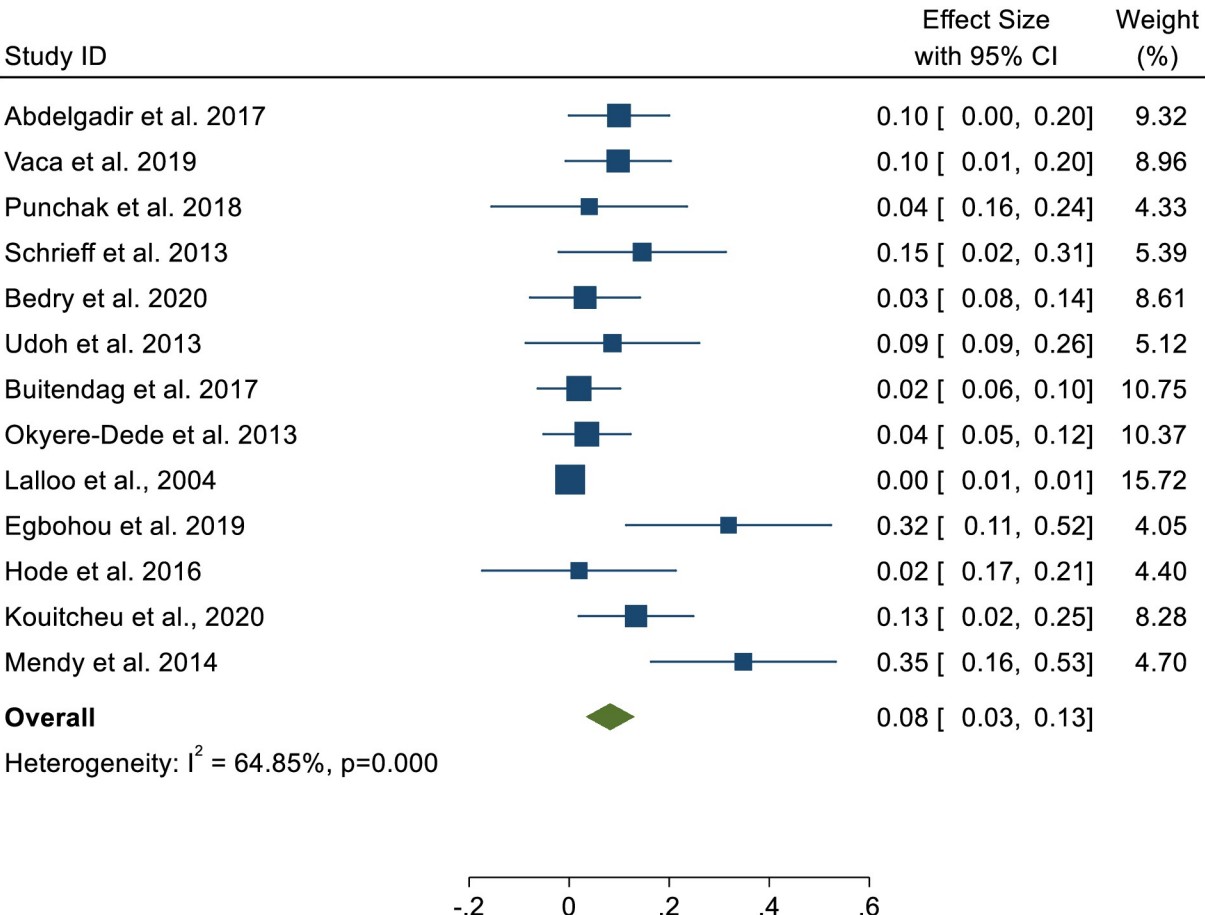

| Study ID | | Effect Size with 95% CI | Weight (%) |
|---|---|---|---|
| Abdelgadir et al. 2017 | | 0.10 [ 0.00, 0.20] | 9.32 |
| Vaca et al. 2019 | | 0.10 [ 0.01, 0.20] | 8.96 |
| Punchak et al. 2018 | | 0.04 [ 0.16, 0.24] | 4.33 |
| Schrieff et al. 2013 | | 0.15 [ 0.02, 0.31] | 5.39 |
| Bedry et al. 2020 | | 0.03 [ 0.08, 0.14] | 8.61 |
| Udoh et al. 2013 | | 0.09 [ 0.09, 0.26] | 5.12 |
| Buitendag et al. 2017 | | 0.02 [ 0.06, 0.10] | 10.75 |
| Okyere-Dede et al. 2013 | | 0.04 [ 0.05, 0.12] | 10.37 |
| Lalloo et al., 2004 | | 0.00 [ 0.01, 0.01] | 15.72 |
| Egbohou et al. 2019 | | 0.32 [ 0.11, 0.52] | 4.05 |
| Hode et al. 2016 | | 0.02 [ 0.17, 0.21] | 4.40 |
| Kouitcheu et al., 2020 | | 0.13 [ 0.02, 0.25] | 8.28 |
| Mendy et al. 2014 | | 0.35 [ 0.16, 0.53] | 4.70 |
| **Overall** | | 0.08 [ 0.03, 0.13] | |

Heterogeneity: $I^2$ = 64.85%, p=0.000

Note: Weight are from Random-effects REML model

**Fig 2. Forest plot of pooled case fatality rate of children's TBI in Sub-Saharan Africa.**

### Risk of bias and sensitivity analysis evaluation

Using the modified Newcastle-Ottawa scale (NOS), we ascertained that three studies [4, 23, 30] had low risk of bias, nine studies [17, 12, 24–29, 32] had a moderate risk of bias, and one study [12] had high risk of bias (S3 Table). A sensitivity analysis was conducted using the random-effects model to verify the impact of individual studies on the pooled case fatality rate of children's TBI in SSA. The findings showed that, there is no influential study on the pooled case fatality rate. The pooled estimated case fatality rate ranged from 7.0%, [95% CI: 2.0%-11.0%] to 9.0%, [95% CI: 5.0%-14.0%] (S4 Table).

### Publication bias

The asymmetrical distribution of funnel plot (Figs 5 and 6) revealed a publication bias among the included studies in the case fatality rate estimate. Similarly, Egger's test yielded statistically significant findings demonstrating the existence of publication bias [p≤0.000]. As a result, Trim-and-fill analysis was used to estimate the number of missing studies that may occur in order to minimize and adjust publication bias in the studies. One study was imputed and approximate pooled case fatality rate was 8.2%, [95% CI:3.4%-13.0%].

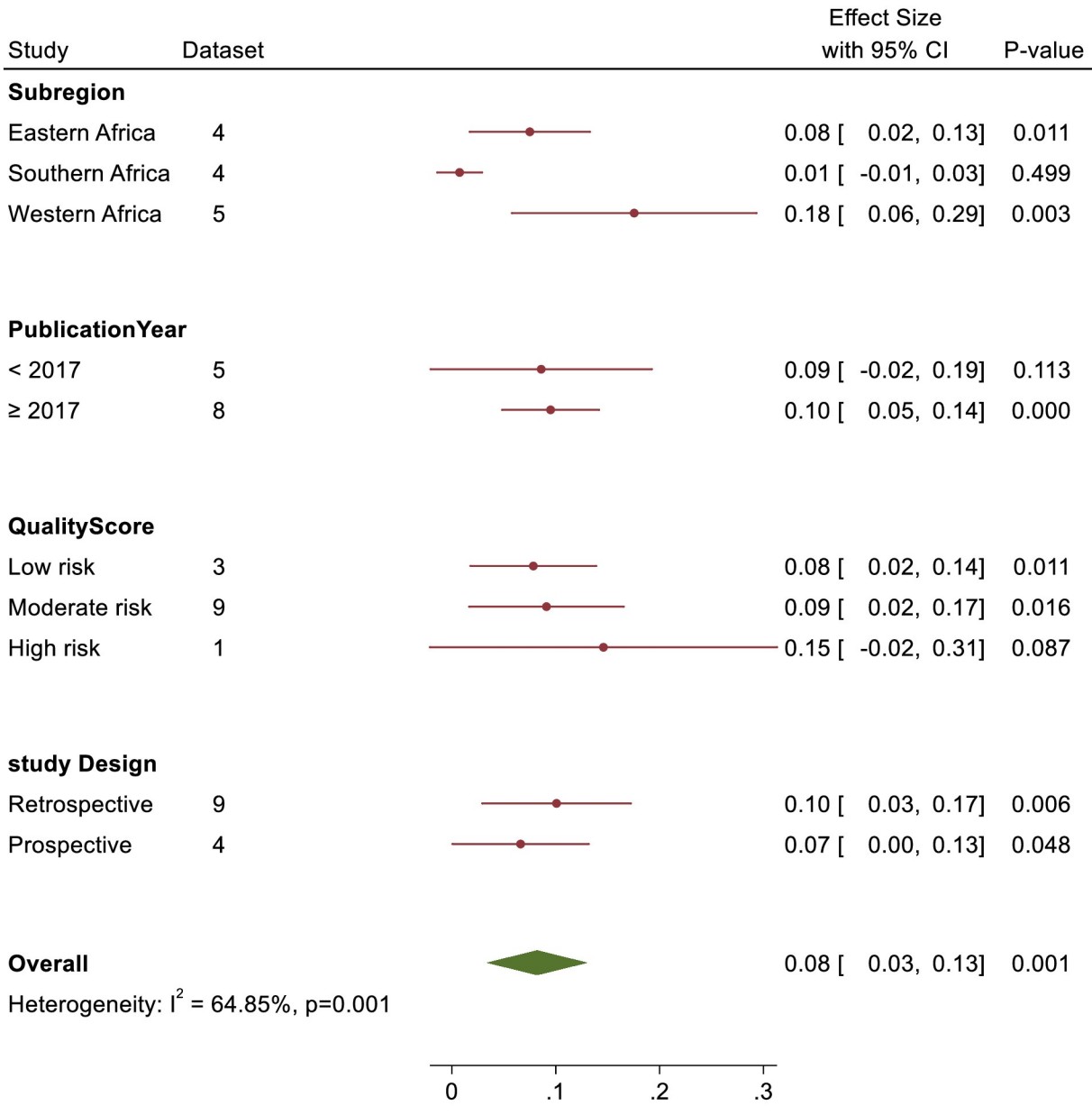

**Fig 3. Forest plot of sub-group analysis of TBI case fatality rate in Sub-Saharan Africa.**

## Discussion

The study aimed to assess the pooled case fatality rate and incidence proportion of pediatrics' TBI and mechanism of injury in SSA. Overall, the incidence proportion and case fatality rate of childhood's TBI were pooled from 4 and 13 studies in SSA respectively.

Our pooled analysis showed that the overall case fatality rate for children's TBI in SSA was 8.0%; [95% CI: 3.0%-13.0%]. and the approximate case fatality rate was adjusted to 8.2%, [95% CI:3.4%-13.0%] after the trim-and-fill analysis was used to correct for publication bias. This approximately corroborates with a study in UK major trauma center with 9.0% reported case

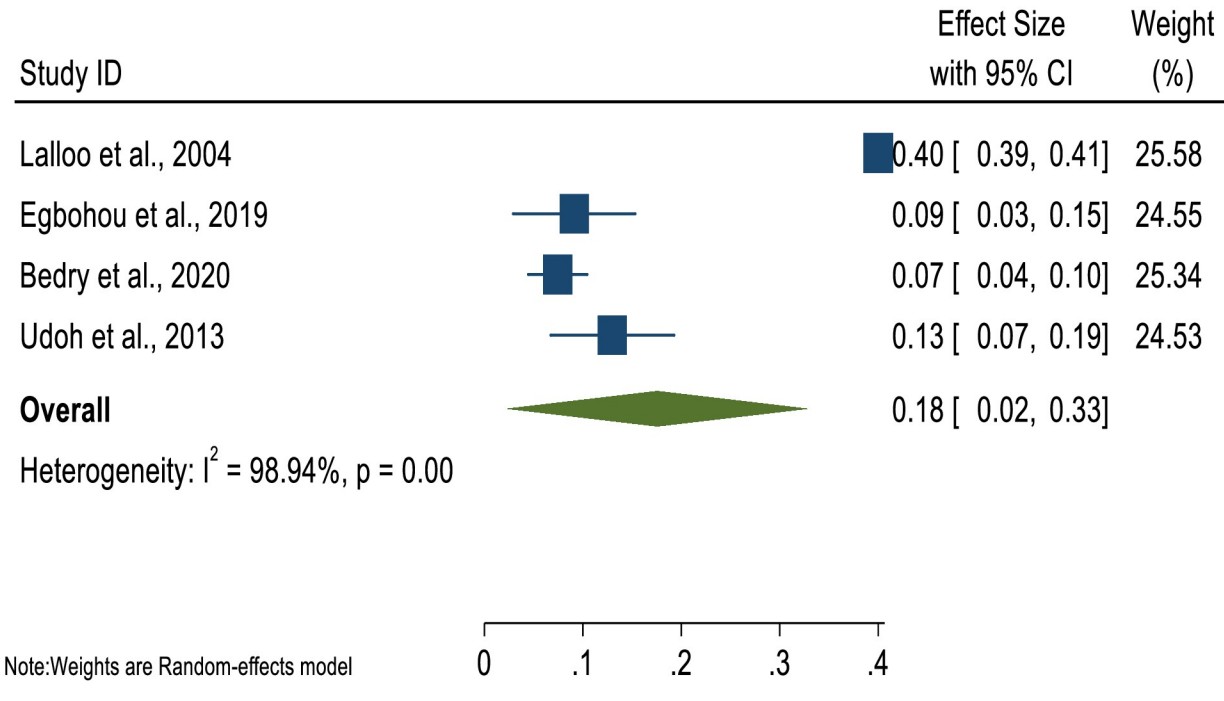

**Fig 4. Forest plot of pooled incidence proportion of children in Sub-Saharan Africa.**

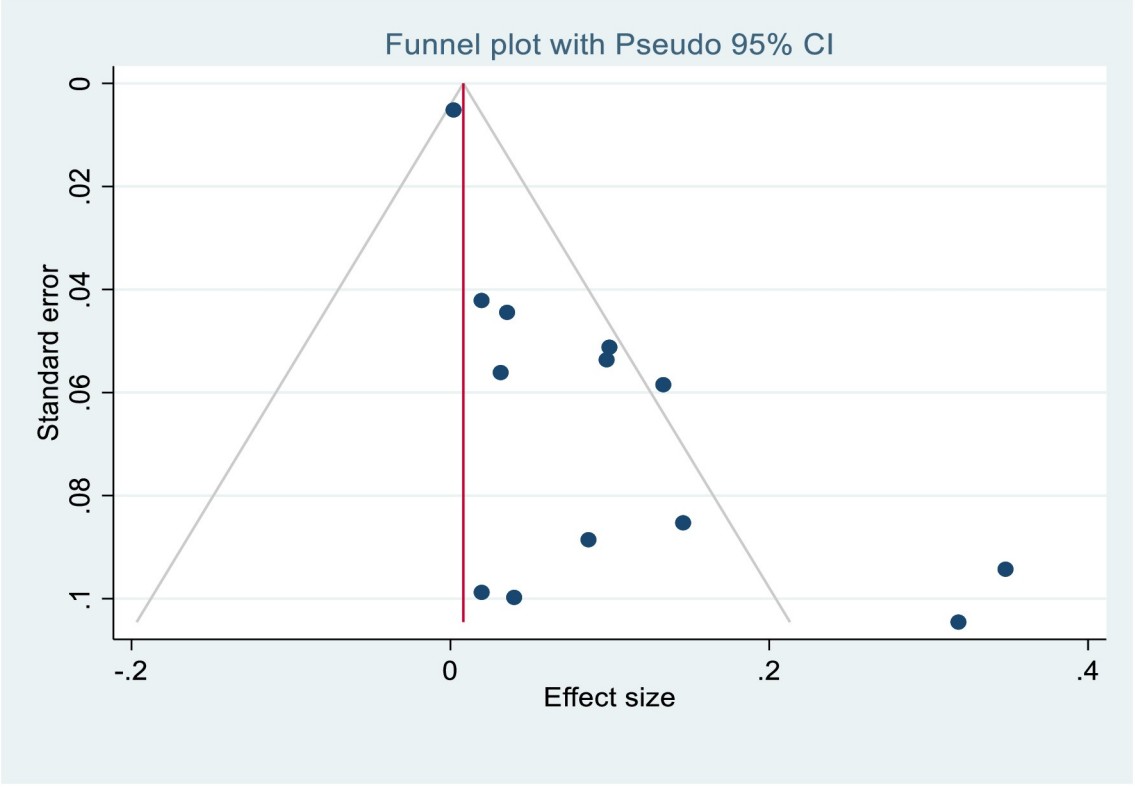

**Fig 5. Funnel plot before Duval's trim and fill analysis.**

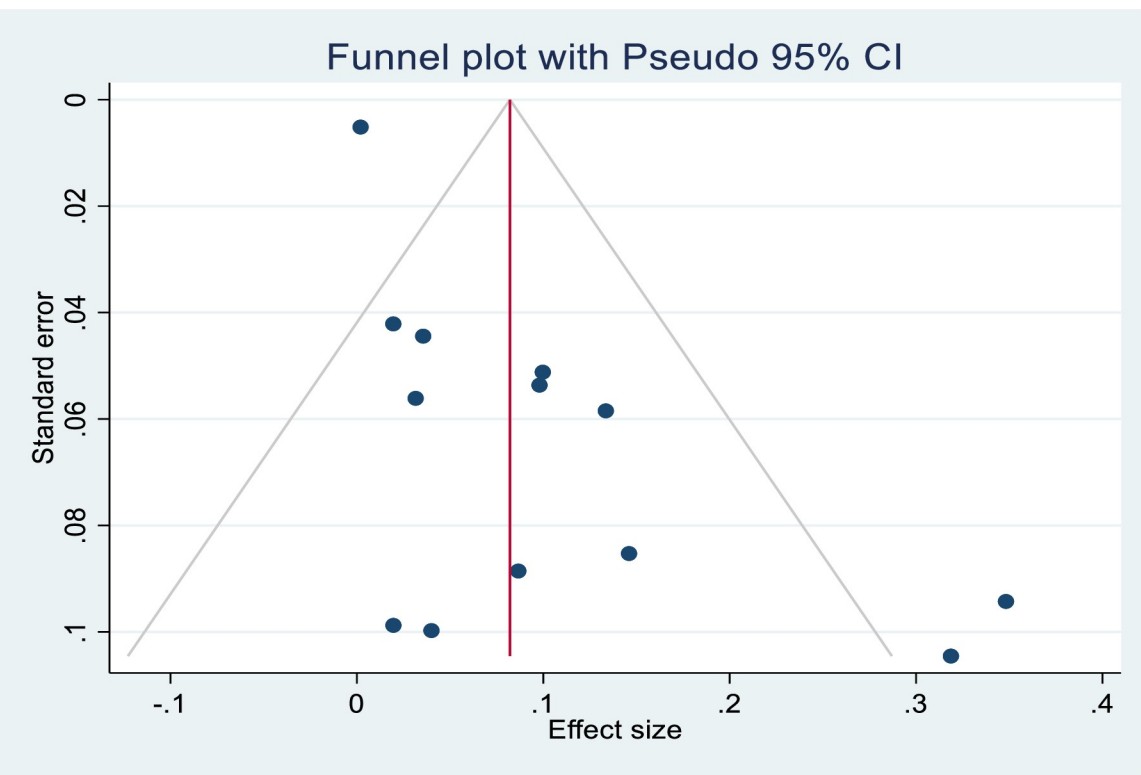

**Fig 6. Funnel plot after Duval's trim and fill analysis.**

fatality [5]. The estimated case fatality rate is higher than those reported in Europe [33], India [34], Australia [35] and United States [36], which reported 3.0%, 3.0% 0.87% and 4.5% respectively. However, our estimate is lower than the 22.8% reported in the US trauma registry [37]. This is not surprising as the study using US trauma registry used only severe children with TBI cases and hence expected that mortality should be high. In fact, studies have identified a strong correlation between severe TBI and in-hospital mortality [30, 31, 38]. The high case fatality in SSA could be ascribed to a variety of factors, including the high severity cases measured by GCS [i.e., 7.7% -40.2%] seen in the current review, infrastructural gap in pre-hospital and in-patients' management that exist in the sub-region as well as the limited specialized Intensive Care Unit for neuro-surgical cases to manage high severe pediatric with TBI in SSA. Our findings suggest that case fatality attributed to children with TBI is of a public health concern in SSA and a well-coordinated effort is needed to curb this menace. As a result, education and prevention, as well as stringent road control measures, must be prioritized.

There was significant variation within the sub-region with highest and lowest case fatality rate occurring in West Africa (18%) and Southern Africa (1%). The wide disparity in case fatality rates between South Africa and West Africa could be linked to late presentation of acute TBI to health facilities, unmet pediatric critical care needs, such as a lack of pediatric Intensive Care Units (ICU) and beds, and insufficiently trained staff in West Africa. For example, in 2018, research found that just one public hospital out of seven has a dedicated ICU, resulting in an estimated 0.4 ICU bed per 100,000 people in Gambia [39]. Siaw Frimpong and colleagues estimated that the critical care capacity was 0.5 ICU beds per 100,000 people in Ghana [40]. Abiodun et al., concluded that there is low survival rate of critically ill children in Nigeria, and as a result training and improved pediatric critical care services and facilities are urgently

needed [41]. Recategorization of the studies into year of publication showed that Children's TBI case fatality is slightly increasing in SSA (i.e., 9% for before 2017 and 10% for studies from 2017 and above). Prospective studies had a low case fatality rate than retrospective studies, according to the research design. In terms of quality score analysis, studies with a high risk of bias had a higher case fatality rate than studies with a low to moderate risk of bias. Just one study was found to have a high probability of bias, which may explain its high case fatality rate.

In the meta-analysis, the pooled incidence proportion of children with TBI in SSA was reported to be 18% [95% CI: 2.0%-33.0%]. However, there was substantial heterogeneity among the studies [$I^2$ = 98.9%, P<0.000]. The current results are similar to a recent study in Qatar (17.7%) [42]. However, our estimate is much higher compared with studies reporting 2.5%(95% CI, 2.3%-2.7%) and 70 cases per 100 000 children in US [43, 44]. Additionally, the global estimate of 50 cases per 100000 per year is lower compared with our estimate [10]. Furthermore, our reported estimate is lower than Alhabdan et al. [45] and Madaan et al. [34] estimates. Our review has also pointed out significant sex differences, as consistently noted that TBI affects male children more often than females in SSA. This finding is in line with several studies findings in different settings [5, 10, 34, 42, 45, 46].

RTA and falls accounted for between 19.1% to 79.1% and 5.1% to 41.2% respectively in this review. Ninety-two percent (92%) of the included studies reported RTA as the leading cause of pediatrics' TBI except Lalloo et al. [26] pointing out fall as the predominant the cause of pediatrics' TBI. Lalloo et al. [26] estimated that more 60% of the injuries occur in child's home environment. The high prevalence of RTA in SSA may be resulting from several vehicular activities taking place as a result of rapid urbanization, as automobiles, bicyclists, pedestrians, and other modes of transportation sharing same highways [7]. Dewan et al. [10] by the same token reported that majority of injuries were caused by RTA and falls. Pedestrians were the most frequent victims of RTAs in Africa and Asia, while vehicle occupants were more common in Australia, Europe, and the United States [10]. As a result, education and prevention, as well as stringent road control measures, must be prioritized in SSA.

## Strength and limitation

Despite the fact that we used a comprehensive search strategy, we restricted ourselves to English-language publications due to a lack of resources, possibly introducing publication bias. In addition, there was significant heterogeneity among the studies. Regardless, this is the first systematic review and meta-analysis on incidence and case fatality rate of TBI among Children (0–18 years) in SSA.

## Conclusion

This is the first systematic review and meta-analysis to assess pooled case fatality rate and incidence proportion of pediatrics' TBI and mechanism of injury in SSA to the best of our knowledge. TBI affects 18% of children aged 0 to 18 years, with almost one-tenth dying in SSA. The most common causes of TBI among this population in SSA were; RTA and falls. TBI incidence and case fatality rate of people aged 0–18 years could be significantly reduced if novel policies focusing on reducing RTA and falls are introduced and implemented in SSA.

## Supporting information

**S1 Table. Preferred Reporting Items for Systematic review and Meta-analyses (PRISMA) checklist.**
(DOC)

**S2 Table. Search strategy for the databases.**
(DOCX)

**S3 Table. Risk of bias assessment.**
(DOCX)

**S4 Table. Leave one out sensitivity analysis.**
(DOCX)

## Acknowledgments

We would like to express our gratitude to all who contributed to the writing of the reviewed articles in this systematic review and meta-analysis. The authors also thank Dr. Louise Ameyaw for extensively proofreading the manuscript.

## Author Contributions

**Conceptualization:** Martin Ackah.

**Data curation:** Martin Ackah, Mohammed Gazali Salifu.

**Formal analysis:** Martin Ackah.

**Investigation:** Martin Ackah, Mohammed Gazali Salifu.

**Software:** Martin Ackah.

**Supervision:** Martin Ackah.

**Validation:** Martin Ackah.

**Visualization:** Martin Ackah.

**Writing – original draft:** Martin Ackah, Mohammed Gazali Salifu, Cynthia Osei Yeboah.

**Writing – review & editing:** Martin Ackah, Mohammed Gazali Salifu, Cynthia Osei Yeboah.

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
