## [Decision Letter · Decision Letter 0]

25 Sep 2021

PONE-D-21-16070Estimated incidence and Mortality Rate of Traumatic Brain Injury among Children (0-18 years) in Sub-Saharan Africa. A Systematic and Meta-Analysis.PLOS ONE

Dear Dr. Ackah,

Thank you for submitting your manuscript to PLOS ONE. After careful consideration, we feel that it has merit but does not fully meet PLOS ONE’s publication criteria as it currently stands. Therefore, we invite you to submit a revised version of the manuscript that addresses the points raised during the review process. The reviewers raised several important issues that must be addressed before this paper can be considered for publication. In addition, please have the manuscript reviewed by an individual with expertise in written English before submitting the revision.

We look forward to receiving your revised manuscript.

Kind regards,

Richard Bruce Mink

Academic Editor

PLOS ONE

Journal Requirements:

2. Please include a separate caption for each figure in your manuscript.

Reviewers' comments:

Reviewer's Responses to Questions

**Comments to the Author**

1. Is the manuscript technically sound, and do the data support the conclusions?

Reviewer #1: Yes

Reviewer #2: Yes

Reviewer #3: Partly

2. Has the statistical analysis been performed appropriately and rigorously? 

Reviewer #1: I Don't Know

Reviewer #2: I Don't Know

Reviewer #3: Yes

3. Have the authors made all data underlying the findings in their manuscript fully available?

Reviewer #1: Yes

Reviewer #2: No

Reviewer #3: Yes

4. Is the manuscript presented in an intelligible fashion and written in standard English?

Reviewer #1: Yes

Reviewer #2: Yes

Reviewer #3: No

5. Review Comments to the Author

Reviewer #1: This manuscript presents a systematic review and meta-analysis to determine incidence and mortality rate in TBI for children in Sub-Saharan Africa (SSA). A total of 13 hospital-based articles were included, reporting on a total of 40687 children with TBI. Pooled mortality rate was 8% with substantial variation across regions. The pooled “incidence proportion of TBI” is reported at 18%.

Whilst I am very happy to see a manuscript on TBI originating from SSA (grossly under-represented in the TBI literature), there are a couple of issues which should be addressed:

1.: Please report your age definition for a pediatric population also in the abstract.

2.: It may be due to my ignorance (in which case I beg forgiveness), but what do you mean by a “pooled incidence proportion of TBI”? Proportion of what? Of all patients seen with TBI? Please clarify.

3.: The review reports on a total of 40687 pediatric patients reported in 13 manuscripts. However, I note that the majority of these come from a single study (92%: Lalloo et al 2004). How did you deal with the over-representation of this study? I further note that in terms of incidence, this study appears to be a substantial outlier (Fig 1)

4.: The heterogeneity between studies is large – Is it then appropriate/permissible to do a meta-analysis?

5.: The mortality rate you report is really a Case fatality rate, being only based on hospital series. Is there any way you could put this in perspective to population-based mortality rates?

Reviewer #2: Thank you fort he opportunity to review this important paper. Although I really can grasp the conclusions and recognise the importance of signaling and preventing traumatic Brain Injuries, I have some comments on the paper.

1. Abstract (and also in Methods) : explain abbreviation AJOL

Page 3: Introduction:

Non degenerative injuries tot he head region: this is a confusing term. Better name it Aquired Brain Injury following trauma. In general - to stress the importance of prevention of TBI- I miss an alinea on the burden of ABI in children who survive, in terms of the phenomenon of growing into deficit with increasing cognitive problems as they grow up and as an important cause of lack/ diminishment of future opportunities or down right dependency in adulthood / contributing cause of poverty.

Page 7: Study characteristics:

It would help to explain in which setting data were obttained in different studies. University/ general large or smaller hospitals?? Any idea how many of the children in the different studies were classified as mild, intermediate or severe TBI?? Do for example children with mild TBI in western SSC reach a hospital/ are counted?? This could really chance the estimated numbers of this study.

At what moment was the GCS measured? Admission tot he hospital/ the lowest measured/ at discharge?? GCS is mentioned but the relevance is not further discussed in the paper.

Page 8: Table 1

I miss in the column “duration” the correct year of the reference of Schrieff et al

In the result section there is mention of : Error! Reference source not found - several times: I presume this is an error itself??

In the discussion I miss discussion of factors as: availability of hospitals with neurosurgical and / or intensive care facilities / organisation of health care/ availability of facilities in general as a factor contributing tot the high mortality. There is a dramatic difference between mortality rates in South Africa and West African SSC. There maybe more contributing factors than a chaotic traffic situation causing this difference

Reviewer #3: The authors present results from a systematic review and meta-analysis of TBI among children in Sub-Saharan Africa. They identified 13 studies that reported on mortality from TBI and 4 studies that included information on incident TBI. They further summarize results across studies on the mechanism of injury. The manuscript will be strengthened if the authors consider the following points:

1. Authors are encouraged to have the manuscript read by a native English speaker as there are numerous places where words are missing, phrasing is awkward, or grammar is incorrect. Examples include "children Traumatic Brain Injury" (in Abstract and elsewhere in manuscript), "10-15 age group 37%" (page 3), "concluded that majority of pediatric" (page 3), "children population" (page 4), "information were" (page 5), sentence starting with "Keywords such as" (page 5 -note there are also some missing quotation marks around the words and missing commas between the words), "broad perspectives parameter" (page 6), "studies into three" (page 6), "articles were remained after duplicate removed" (page 7), "The current reviewed showed that" (page 7), and "However, lower than reported in US trauma registry 22.8%" (page 11). Authors also use capitalization unnecessarily ("in Children" (page 3), "whereas Road traffic accidents" (page 3), "severe Pediatric TBI" (page 11), "that Children's TBI" (page 11)) or do not use it when it should be used ("west Africa" (page 2 and 9), "United states" (page 11), "and Mortality Rate" and "among Children" (page 13))

2. Table 1 should include the number of deaths for each study.

3. On page 6, authors state that the average NOS score from two reviewers was used as the final score. Were there any major differences between the reviewers?

4. On page 7 (and in the Abstract and Discussion), when authors present the percentages of RTA and Falls across studies, authors should clarify they are presenting the range of observed percentages, so readers don't think this is a confidence interval or some other estimated quantity.

5. Authors do not refer to any of the figures within the text of the manuscript.

6. Figure 1: authors should provide reasons for exclusion for the box of 200 that gets reduced to 70.

7. On page 9, authors talk about subgroup analyses related to levels of risk of bias, but authors have not yet summarized the studies according to risk of bias (that comes on page 10). Authors might consider reporting on the summary of the risk of bias earlier in the results, so the subgroup analysis has some context.

8. Figure 3 - there is a typo "Moderate Rrisk". Authors should also define in the figure caption what the p-value represents.

9. When authors report the sensitivity analysis of the estimated mortality by leaving one study out, they cite specific studies (page 10). They should be clear in the text that these cited references are the ones left out of the estimate. Also, authors refer to 4 studies for the 9% estimate. There are 4 different studies that when left out, the estimate is 9%, but the confidence interval is not the same across those 4 results. Authors might consider not citing the studies in the sentence, since readers can refer to the table to see which studies were removed for the different results.

10. In the discussion (page 12), authors refer to results about sex differences, but these were not presented in the manuscript.

Minor points:

1. Authors redefine acronyms multiple times (TBI is defined three times within the Abstract, LMIC is defined twice on page 3). Acronyms or abbreviations only need to be defined once in the Abstract (if used) and then once in the main body of the manuscript. Authors do not define AJOL, which appears both in the Abstract and in the body of the manuscript.

2. Throughout the manuscript (Abstract and main text), authors refer to a total of 40687 children across the 13 studies, but with the numbers provided in Table 1, the total number appears to only be 40685. Authors should carefully check the numbers and correct where needed.

3. on page 5, authors state the search was limited to 2000-December 2020. To be clear, authors should state January 2000 - December 2020.

4. on page 5, authors refer to S2 Table 2, but I believe it is S2 Table 1.

5. on page 7: "Glasgow Come Scale" should be "Glasgow Coma Scale"

6. There are several places in the manuscript with "Error! Reference source not found". Authors should check their references.

7. page 10: authors state there are 10 studies of moderate risk of bias, but this should be 9 (based on the cited references and Figure 3).

8. page 10: authors refer to S2 Table 2, but the file is called S3 Table 2. Also, in the table, the Egbonhou study does not have the risk level filled in.

9. Authors define RTA to be road traffic accidents, but then appear to switch to using RTI (maybe road traffic incidents) in the Discussion. Authors should be consistent in their terminology.

10. Figure captions all say "Figure 1"

6. PLOS authors have the option to publish the peer review history of their article (what does this mean?). If published, this will include your full peer review and any attached files.

Reviewer #1: **Yes: **Andrew I.R. Maas

Reviewer #2: No

Reviewer #3: No

---

## [Author Response · Author response to Decision Letter 0]

1 Oct 2021

Response to reviewers’ comments

I sincerely express my warmest greetings to you and your prestigious journal for your comments and feedback. I write on behalf of my co-authors to submit our reply to your astute experienced reviewers' insightful comments. The methodology used follows a point-by-point approach to responding to all comments. Please see below for our response.

Reviewer 1

Comment: Please report your age definition for a pediatric population also in the abstract

Authors’ response: Authors have taken the reviewer’s comment into consideration and accordingly report the age definition for pediatric population in the abstract. ‘’ Specifically, the modified portion now reads “reliable estimate of incidence and outcome of children (0-18 years) with TBI”

Comment: It may be due to my ignorance (in which case I beg forgiveness), but what do you mean by a “pooled incidence proportion of TBI”? Proportion of what? Of all patients seen with TBI? Please clarify

Authors’ response: The pooled incidence proportion in this situation refers to all head injuries reported to the emergency department [ED] that were children with TBI in SSA. This is measured in % as compared to incident rate which is measured in persons years.

Comment: The review reports on a total of 40687 pediatric patients reported in 13 manuscripts. However, I note that the majority of these come from a single study (92%: Lalloo et al 2004). How did you deal with the over-representation of this study? I further note that in terms of incidence, this study appears to be a substantial outlier (Figure 1)

Authors’ response: Thanks for the comment. This situation was investigated through sensitivity analysis and the results showed that none of the studies had significant impact on the outcomes [page 10]. Also, in the pooled incidence figure the ‘’weight’’ given to Lalloo et al 2004 was 25.58% which is not significantly different from the weight from other studies [i.e., 24.55%, 25.34%, and 24.53%]

Comment: The heterogeneity between studies is large – Is it then appropriate/permissible to do a meta-analysis?

Authors’ response: The large heterogeneity was adjusted during the meta-analysis using DerSimonian and Laird random-effects model (REM) at 95% confidence interval.

Comment: The mortality rate you report is really a Case fatality rate, being only based on hospital series. Is there any way you could put this in perspective to population-based mortality rates?

Authors’ response: Authors have taken the reviewer’s comment into consideration and accordingly modified the topic to incorporate case-fatality rates and as a result ‘’mortality’’ changed to ‘’case-fatality rate’’ throughout the manuscript. The topic now reads ‘’ Estimated incidence and Case Fatality Rate of Traumatic Brain Injury among Children (0-18 years) in Sub-Saharan Africa. A Systematic review and Meta-Analysis

Reviewer 2

Comment: Abstract (and also in Methods): explain abbreviation AJOL

Authors’ response: Authors acknowledge the relevance of reviewer’s comment and as a result, AJOL abbreviation has been explained in the abstract and also in the method [i.e., AJOL=African Journal Online]

Comment: Page 3 Introduction- Non degenerative injuries to the head region: this is a confusing term. Better name it Acquired Brain Injury following trauma

Authors’ response: Authors have taken the reviewer’s comment into consideration and accordingly modified the sentence in Page 3. The sentence now reads ‘’Traumatic Brain Injury (TBI) in Children is acquired brain injuries following trauma, and is similar to those of adults but differs in both management and pathophysiology’’

Comment: In general - to stress the importance of prevention of TBI- I miss a line on the burden of ABI in children who survive, in terms of the phenomenon of growing into deficit with increasing cognitive problems as they grow up and as an important cause of lack/ diminishment of future opportunities or down right dependency in adulthood / contributing cause of poverty

Authors’ response: Authors acknowledge the relevance of reviewer’s comment and as a result added a paragraph which reads ‘’Berger et al estimated that only 65% of children with severe TBI survive (Edwards & Bartkowski, 1985; Schrieff, Thomas, Dollman, Rohlwink, & Figaji, 2013). The outcome of brain injury is very detrimental to the child, family and by extension the country. For instance, studies have found that TBI can lead to long-term cognitive and neurobehavioral deficiencies, as well as intellectual, academic, and personality adjustment issues, and familial stress (Hawley, Ward, Magnay, & Long, 2004; Klonoff, Clark, & Klonoff, 1993). This could lead to a reduction in future possibilities or outright dependency in adulthood, both of which are contributing factors to poverty’’

Comment: It would help to explain in which setting data were obtained in different studies. University/ general large or smaller hospitals??

Authors’ response: Authors have taken the reviewer’s comment into consideration and accordingly added the different study settings [Tabe 1]

Comment: Any idea how many of the children in the different studies were classified as mild, intermediate or severe TBI??

Authors’ response: The authors have now extracted and added TBI severity classification in terms of mild, moderate or severe TBI [Table 1]

Comment: At what moment was the GCS measured?

Authors’ response: Thanks for the comment. All the GCS in the studies were measured at Admission. This has further been clarified in Table 1

Comment: Table 8- I miss in the column “duration” the correct year of the reference of Schrieff et

Authors’ Response: The correct duration year has been rectified. The column now reads; 2000-2011

Comment: In the result section there is mention of: Error! Reference source not found - several times: I presume this is an error itself??

Authors’ response: Thanks for the comment. It an editorial and software issues during the submission. This has been rectified.

Comment: In the discussion I miss discussion of factors as: availability of hospitals with neurosurgical and / or intensive care facilities / organization of health care/ availability of facilities in general as a factor contributing to the high mortality

Authors response: The authors have now discussed the point raised. The sentence reads ‘The high case fatality in SSA could be ascribed to a variety of factors, including the high severity cases measured by GCS [i.e., 7.7% -40.2 %] seen in the current review, infrastructural gap in pre-hospital and in-patients’ management that exist in the sub-region as well as the limited specialized Intensive Care Unit for neuro-surgical cases to manage high severe pediatric with TBI in SSA’’ [Page 11]

Comment: There is a dramatic difference between mortality rates in South Africa and West African SSC. There may be more contributing factors than a chaotic traffic situation causing this difference

Authors response: Authors acknowledge the relevance of reviewer’s comment and as a result added additional possible factors contributing to the high case fatality in West Africa. It reads’’ The wide disparity in case fatality rates between South Africa and West Africa could be linked to late presentation of acute TBI to health facilities, unmet pediatric critical care needs, such as a lack of pediatric Intensive Care Units (ICU) and beds, and insufficiently trained staff in West Africa. For example, in 2018, research found that just one public hospital out of seven has a dedicated ICU, resulting in an estimated 0.4 ICU bed per 100,000 people in Gambia40. Siaw Frimpong and colleagues estimated that the critical care capacity was 0.5 ICU beds per 100,000 people in Ghana41. Abiodun et al., concluded that there is low survival rate of critically ill children in Nigeria, and as a result training and improved pediatric critical care services and facilities are urgently needed 42 [page 11-12]

Reviewer 3

Comment: Authors are encouraged to have the manuscript read by a native English speaker as there are numerous places where words are missing, phrasing is awkward, or grammar is incorrect. Examples include "children Traumatic Brain Injury" (in Abstract and elsewhere in manuscript), "10-15 age group 37%" (page 3), "concluded that majority of pediatric" (page 3), "children population" (page 4), "information were" (page 5), sentence starting with "Keywords such as" (page 5 -note there are also some missing quotation marks around the words and missing commas between the words), "broad perspectives parameter" (page 6), "studies into three" (page 6), "articles were remained after duplicate removed" (page 7), "The current reviewed showed that" (page 7), and "However, lower than reported in US trauma registry 22.8%" (page 11). Authors also use capitalization unnecessarily ("in Children" (page 3), "whereas Road traffic accidents" (page 3), "severe Pediatric TBI" (page 11), "that Children's TBI" (page 11)) or do not use it when it should be used ("west Africa" (page 2 and 9), "United states" (page 11), "and Mortality Rate" and "among Children" (page 13))

Authors’ response: As recommended by the reviewer, the entire manuscript has been thoroughly read once again by all authors and a third independent editor to correct all grammatical errors which addresses the grammatical concerns highlighted by the reviewer.

Comment: Table 1 should include the number of deaths for each study.

Authors’ response: Authors have taken the reviewer’s comment into consideration and accordingly added the number of deaths for each study in Table 1.

Comment: On page 6, authors state that the average NOS score from two reviewers was used the final score. Were there any major differences between the reviewers?

Authors response: Thanks for the comment. There was no major difference between the reviewers. The inter-rater reliability was 0.9 [kappa=0.9]. This has been added to the manuscript for clarity. [Page 6]

Comment: On page 7 (and in the Abstract and Discussion), when authors present the percentages of RTA and Falls across studies, authors should clarify they are presenting the range of observed percentages, so readers don't think this is a confidence interval or some other estimated quantity.

Authors’ response: Authors have taken the reviewer’s comment into consideration and accordingly rewritten the statements to avoid confusion and ambiguity both in the abstract and manuscript [Page 7]

Comment: Authors do not refer to any of the figures within the text of the manuscript.

Authors’ response: Thanks for the comment. Authors referred however, during submission this came as an error during the submission. The authors believe that this will be resolved or rectify at the editorial level. 

Comment: Figure 1: authors should provide reasons for exclusion for the box of 200 that gets reduced to 70.

Authors’ response: Thanks for the comment. They were all duplicate articles which has been explained in the second box 

Comment: On page 9, authors talk about subgroup analyses related to levels of risk of bias, but authors have not yet summarized the studies according to risk of bias (that comes on page 10). Authors might consider reporting on the summary of the risk of bias earlier in the results, so the subgroup analysis has some context

Authors’ response: Thanks for the comment. The risk of bias assessment was provided and summarized in the supplementary file [S3 Table 2].

Comment: Figure 3 - there is a typo "Moderate Rrisk". Authors should also define in the figure caption what the p-value represents.

Authors’ response: Thanks for the comment. The typo error has been rectified 

Comment: When authors report the sensitivity analysis of the estimated mortality by leaving one study out, they cite specific studies (page 10). They should be clear in the text that these cited references are the ones left out of the estimate. Also, authors refer to 4 studies for the 9% estimate. There are 4 different studies that when left out, the estimate is 9%, but the confidence interval is not the same across those 4 results. Authors might consider not citing the studies in the sentence, since readers can refer to the table to see which studies were removed for the different results.

Authors’ response: Thanks for the comment. Authors have now removed the cited references to prevent any confusion and ambiguity 

Comment: In the discussion (page 12), authors refer to results about sex differences, but these were not presented in the manuscript

Authors’ response: Thanks for the comment. The sex ratio was reported in Table 1

Minor points

Comment: Authors redefine acronyms multiple times (TBI is defined three times within the Abstract, LMIC is defined twice on page 3). Acronyms or abbreviations only need to be defined once in the Abstract (if used) and then once in the main body of the manuscript. Authors do not define AJOL, which appears both in the Abstract and in the body of the manuscript.

Authors’ response: Thanks for the comments. This has now been rectified throughout the manuscript

Comment: Throughout the manuscript (Abstract and main text), authors refer to a total of 40687 children across the 13 studies, but with the numbers provided in Table 1, the total number appears to only be 40685. Authors should carefully check the numbers and correct where needed.

Authors’ response: Thanks for the comment. Authors have carefully checked and corrected this anomaly. The total sample is 40685.

Comment: on page 5, authors state the search was limited to 2000-December 2020. To be clear, authors should state January 2000 - December 2020

Authors’ response: Thanks for the comment. Authors have resolved it. It now reads ‘’ The search was limited to January, 2000- December, 2020’’

Comment: on page 5, authors refer to S2 Table 2, but I believe it is S2 Table 1

Authors’ response: Thanks for the comment. Authors have now renamed the file as S2 Table [Page 6]

Comment: on page 7: "Glasgow Come Scale" should be "Glasgow Coma Scale"

Authors response: Thanks for the comment. Authors have now corrected this mistake [page 9)

Comment: There are several places in the manuscript with "Error! Reference source not found". Authors should check their references

Authors’ response: Thanks for the comment: This has been rectified.

Comment: page 10: authors state there are 10 studies of moderate risk of bias, but this should be 9 (based on the cited references and Figure 3).

Authors’ response: Thanks for comment. This has been rectified. There were 9 studies with moderate risk of bias in the study.

Comment: page 10: authors refer to S2 Table 2, but the file is called S3 Table 2. Also, in the table, the Egbonhou study does not have the risk level filled in

Authors’ response: Thank for the comment. Authors have now renamed the file as S3 Table. Also, Egbonhou et al risk of bias now filled.

Comment: Authors define RTA to be road traffic accidents, but then appear to switch to using RTI (maybe road traffic incidents) in the Discussion. Authors should be consistent in their terminology.

Authors’ response: Thanks for the comment. Authors have now rectified this inconsistency.

Comment: Figure captions all say "Figure 1"

Authors response: Thanks for the comment. This is a software processing. The authors believe this will be solved editorially.

.

---

## [Decision Letter · Decision Letter 1]

18 Nov 2021

PONE-D-21-16070R1Estimated incidence and Case Fatality Rate of Traumatic Brain Injury among Children (0-18 years) in Sub-Saharan Africa. A Systematic Review and Meta-Analysis.PLOS ONE

Dear Dr. Ackah,

Thank you for submitting your manuscript to PLOS ONE. After careful consideration, we feel that it has merit but does not fully meet PLOS ONE’s publication criteria as it currently stands. Therefore, we invite you to submit a revised version of the manuscript that addresses the points raised during the review process.

You have addressed most of the concerns previously raised by the reviewers but there are a few minor issues that still need to be addressed.

We look forward to receiving your revised manuscript.

Kind regards,

Richard Bruce Mink

Academic Editor

PLOS ONE

Journal Requirements:

Reviewers' comments:

Reviewer's Responses to Questions

**Comments to the Author**

1. If the authors have adequately addressed your comments raised in a previous round of review and you feel that this manuscript is now acceptable for publication, you may indicate that here to bypass the “Comments to the Author” section, enter your conflict of interest statement in the “Confidential to Editor” section, and submit your "Accept" recommendation.

Reviewer #1: (No Response)

Reviewer #3: (No Response)

2. Is the manuscript technically sound, and do the data support the conclusions?

Reviewer #1: Yes

Reviewer #3: Yes

3. Has the statistical analysis been performed appropriately and rigorously? 

Reviewer #1: Yes

Reviewer #3: Yes

4. Have the authors made all data underlying the findings in their manuscript fully available?

Reviewer #1: Yes

Reviewer #3: Yes

5. Is the manuscript presented in an intelligible fashion and written in standard English?

Reviewer #1: Yes

Reviewer #3: Yes

6. Review Comments to the Author

Reviewer #1: This manuscript is a revision of a previous submission. The authors have mostly addressed all reviewer comments appropriately. Overall, the manuscript is much improved. Personally, I would not really consider all changes an improvement, but this is not the fault of the authors as changes resulted from specific requests/suggestions of reviewers. There are a few relatively minor issues that remain or have arisen anew following the changes implemented:

1.: Abstract, Results: “The current reviewed” should be “The current review”.

2.: Introduction, line 1: I would suggest to change “acquired brain injuries” to “acquired brain injury”

3.: Results, page 8 at the bottom: The sentence “This also ranged from 5.1% in South Africa 29, and 41.2% in south Africa 26” reads a bit strange. It is strange that the extremes of the range are within one country (South Africa). On looking up the citation, they appear to even be from the same hospital, but over a different time period. I would suggest to either delete the entire sentence, to delete the specific mention of South Africa, or – if maintained – add an explanation.

Reviewer #3: The authors have addressed the majority of my earlier concerns. There remain a few minor points and some grammatical/English edits that should be addressed:

1. Abstract (Introduction section): "need for accurate" should be "need for an accurate"

2. Abstract (Results section):"The current reviewed showed" should be "The current review showed"

3. page 5 (1st line): "as the most cause" should be "as the most common cause" and "in all age group" should be "in all age groups"

4. page 5 (1st full paragraph): "need for accurate and" should be "need for an accurate and"

5. page 8 (Study Selection): "inclusion criteria and included" should be "inclusion criteria and were included"

6. Figure 1: in the authors' response to my comment about this Figure, they said the explanation for the reduction from 200 to 70 was due to duplicate records. Maybe there is just confusion in how to read their figure. Typically, these figures show how many articles are at each stage with an intermediate box that shows how many were excluded (and reasons for exclusion) at each stage. For example, their box for "Records after abstracts and title screened" has n=70 articles and the box to the right says that 40 were excluded which then yields the 30 articles in the next box. My original question had to do with the lack of boxes to the right for the 1st 2 boxes (articles identified through database and record after duplicate removed). So, did authors arrive at 200 articles from the original 820 after removing duplicates? If not, what were the reasons that reduced the number from 820 to 200? Similarly, what were the reasons for reducing the 200 articles to 70 articles?

7. page 9 (2nd to last line): change "from 5.1% in South Africa, and 41.2% in south Africa" to "from 5.1% to 41.2%".

8. page 12: change "This approximately corroborate with" to "This approximately corroborates with", "not surprising as study" to "not surprising as the study", "TBI cases hence" to "TBI cases and hence", and "well-coordinated effort are" to "well-coordinated effort is"

9. page 13: change "The current results is similar" to "The current results are similar"

10. page 13: I previously made a comment about the sex-differences in TBI - these are first mentioned in the Discussion. If authors feel this point is important enough to make in the Discussion, it should also be mentioned in the results.

11. page 14: change "fall predominant the cause" to "fall as the predominant cause" and remove "whooping" from the Conclusion section

7. PLOS authors have the option to publish the peer review history of their article (what does this mean?). If published, this will include your full peer review and any attached files.

Reviewer #1: No

Reviewer #3: No

---

## [Author Response · Author response to Decision Letter 1]

23 Nov 2021

Response to reviewers’ comments

I sincerely express my warmest greetings to you and your prestigious journal for your comments and feedback. I write on behalf of my co-authors to submit our reply to your astute experienced reviewers' insightful comments. The methodology used follows a point-by-point approach to responding to all comments. Please see below for our response.

Reviewer #1:

Comment: Abstract, Results: “The current reviewed” should be “The current review”.

Response: Thanks for the correction. This has been rectified

Comment: Introduction, line 1: I would suggest to change “acquired brain injuries” to “acquired brain injury”

Response: Thanks for the correction. This has been rectified

comment: Results, page 8 at the bottom: The sentence “This also ranged from 5.1% in South Africa 29, and 41.2% in south Africa 26” reads a bit strange. It is strange that the extremes of the range are within one country (South Africa). On looking up the citation, they appear to even be from the same hospital, but over a different time period. I would suggest to either delete the entire sentence, to delete the specific mention of South Africa, or – if maintained – add an explanation.

Response: Thanks for the correction. The sentence has now been deleted.

Reviewer #3 

comment: Abstract (Introduction section): "need for accurate" should be "need for an accurate"

Response: Thanks for the correction. This has been rectified

comment: page 5 (1st line): "as the most cause" should be "as the most common cause" and "in all age group" should be "in all age groups"

response: Thanks for the correction. This has been rectified

comment: page 5 (1st full paragraph): "need for accurate and" should be "need for an accurate and"

response: Thanks for the correction. This has been rectified

comment: page 8 (Study Selection): "inclusion criteria and included" should be "inclusion criteria and were included"

response: Thanks for the correction. This has been rectified

 comment: Figure 1: in the authors' response to my comment about this Figure, they said the explanation for the reduction from 200 to 70 was due to duplicate records. Maybe there is just confusion in how to read their figure. Typically, these figures show how many articles are at each stage with an intermediate box that shows how many were excluded (and reasons for exclusion) at each stage. For example, their box for "Records after abstracts and title screened" has n=70 articles and the box to the right says that 40 were excluded which then yields the 30 articles in the next box. My original question had to do with the lack of boxes to the right for the 1st 2 boxes (articles identified through database and record after duplicate removed). So, did authors arrive at 200 articles from the original 820 after removing duplicates? If not, what were the reasons that reduced the number from 820 to 200? Similarly, what were the reasons for reducing the 200 articles to 70 articles?

Response: Authors acknowledge the relevance of the reviewer’s comment and as a result, revised Figure 1.

Comment: page 9 (2nd to last line): change "from 5.1% in South Africa, and 41.2% in south Africa" to "from 5.1% to 41.2%".

Responses: Thanks for the correction. The sentence has now been deleted as recommended by reviewer 1.

Comment: page 12: change "This approximately corroborate with" to "This approximately corroborates with", "not surprising as study" to "not surprising as the study", "TBI cases hence" to "TBI cases and hence", and "well-coordinated effort are" to "well-coordinated effort is"

response: Thanks for the corrections. These have been rectified

comment: page 13: change "The current results is similar" to "The current results are similar"

comment: page 13: I previously made a comment about the sex-differences in TBI - these are first mentioned in the Discussion. If authors feel this point is important enough to make in the Discussion, it should also be mentioned in the results.

Response: Thanks for the comment. The sex difference has been mentioned in the result section [Page 8]

Comment: page 14: change "fall predominant the cause" to "fall as the predominant cause" and remove "whooping" from the Conclusion section

response: Thanks for the corrections. These have been rectified

---

## [Editor Report · Decision Letter 2]

13 Dec 2021

Estimated incidence and Case Fatality Rate of Traumatic Brain Injury among Children (0-18 years) in Sub-Saharan Africa. A Systematic Review and Meta-Analysis.

PONE-D-21-16070R2

Dear Dr. Ackah,

We’re pleased to inform you that your manuscript has been judged scientifically suitable for publication and will be formally accepted for publication once it meets all outstanding technical requirements.

Kind regards,

Richard Bruce Mink

Academic Editor

PLOS ONE
---

## [Editor Report · Acceptance letter]

15 Dec 2021

PONE-D-21-16070R2 

Estimated incidence and Case Fatality Rate of Traumatic Brain Injury among Children (0-18 years) in Sub-Saharan Africa. A Systematic review and Meta-Analysis. 

Dear Dr. Ackah:

I'm pleased to inform you that your manuscript has been deemed suitable for publication in PLOS ONE. Congratulations! Your manuscript is now with our production department. 

Kind regards, 

on behalf of

Dr. Richard Bruce Mink 

Academic Editor

PLOS ONE